# Quantitative three-dimensional imaging of chemical short-range order via machine learning enhanced atom probe tomography

Yue Li [1] ✉, Ye Wei [1], Zhangwei Wang [2] ✉, Xiaochun Liu [3], Timoteo Colnaghi [4], Liuliu Han [1], Ziyuan Rao [1], Xuyang Zhou [1], Liam Huber[1], Raynol Dsouza[1], Yilun Gong [1], Jörg Neugebauer [1], Andreas Marek [4], Markus Rampp[4], Stefan Bauer [5], Hongxiang Li [6], Ian Baker [7], Leigh T. Stephenson [1] & Baptiste Gault [1,8] ✉

Chemical short-range order (CSRO) refers to atoms of specific elements self-organising within a disordered crystalline matrix to form particular atomic neighbourhoods. CSRO is typically characterized indirectly, using volume-averaged or through projection microscopy techniques that fail to capture the three-dimensional atomistic architectures. Here, we present a machine-learning enhanced approach to break the inherent resolution limits of atom probe tomography enabling three-dimensional imaging of multiple CSROs. We showcase our approach by addressing a long-standing question encountered in body-centred-cubic Fe-Al alloys that see anomalous property changes upon heat treatment. We use it to evidence non-statistical $B_2$-CSRO instead of the generally-expected $D0_3$-CSRO. We introduce quantitative correlations among annealing temperature, CSRO, and nano-hardness and electrical resistivity. Our approach is further validated on modified $D0_3$-CSRO detected in Fe-Ga. The proposed strategy can be generally employed to investigate short/medium/long-range ordering phenomena in different materials and help design future high-performance materials.

Ordering is the preferential occupation of specific sites in crystalline materials by particular elements. The very early stage of thermally-activated ordering is referred to as chemical short-range order (CSRO)[1–3], the occurrence of which has been reported to significantly change the mechanical[4–13] and functional[14–16] performances of materials. Fe-Al alloys, lightweight materials with excellent strength and corrosion/wear resistance, are good candidates for theoretical studies on such ordering transformations[16–18]. Undeformed Fe-Al alloys with compositions close to the boundary of the disorder-order transition,

annealed at a relatively low temperature (e.g. 523 K) and then quenched, often exhibit an anomalous increase in electrical resistivity[14,19,20]. This unusual behaviour results from the so-called K-state initially reported in 1951[14] and generally attributed to $D0_3$-CSRO ($Fe_3Al$-like), seen as the onset of long-range ordering reactions[21,22]. The K-state was reported in a vast array of alloy systems, e.g. Ni-X (X=Cr, Co, Al, Fe), Fe-X (X=Si, Ga, Cr), Cu-X (X=Ni, Mn), Ni-Cu-Zn, Ni-Fe-Cr, Fe-Al-Cr[14,19,20,23,24]. However, the underpinnings of the K-state are still unclear and the quantitative characterization of CSRO remains a formidable challenge.

[1]Max-Planck Institut für Eisenforschung GmbH, Max-Planck-Straße 1, 40237 Düsseldorf, Germany. [2]State Key Laboratory of Powder Metallurgy, Central South University, Changsha 410083, China. [3]Institute of Metals, College of Materials Science and Engineering, Changsha University of Science and Technology, Changsha 410114, China. [4]Max Planck Computing and Data Facility, Gießenbachstraße 2, 85748 Garching, Germany. [5]Max Planck Institute for Intelligent Systems, Max-Planck-Ring 4, 72076 Tübingen, Germany. [6]State Key Laboratory for Advanced Metals and Materials, University of Science and Technology Beijing, 100083 Beijing, China. [7]Thayer School of Engineering, Dartmouth College, Hanover, NH 03755, USA. [8]Department of Materials, Imperial College, South Kensington, London SW7 2AZ, UK. ✉e-mail: yue.li@mpie.de; z.wang@csu.edu.cn; b.gault@mpie.de

As alloys increase in complexity, the knowledge gap linking CSRO to material properties hinders exploiting it in material design to enhance material performance[12,25,26].

CSRO was conventionally characterized by indirect, volume-averaged bulk experimental techniques, such as X-ray or neutron scattering, or Mössbauer spectroscopy[14,16,27,28], leaving much room for interpretation on how CSRO proceeds at the atomic scale. Recently, transmission electron microscopy (TEM)-based approaches characterized CSRO in compositionally-complex alloys based on enhanced diffraction contrast or local compositional mapping[12,26,29,30]. TEM only provides two-dimensional, through-thickness projections of CSRO, leading to limitation in the measurement of size/morphology of CSRO domains, which will be discussed later. With near-atomic spatial resolution and high analytical sensitivity (10–100 ppm), atom probe tomography (APT) offers, in principle, a unique opportunity for quantifying the size and morphology of CSRO domains in three dimensions (3D)[31,32]. However, its spatial resolution prevents it from precisely imaging atomic ordering processes, particularly domains below 1 nm in radius[3,31,33,34].

Here, to overcome this inherent limitation and unveil CSRO, we propose a machine learning (ML) enhanced approach based on a convolutional neural network (CNN) applied to spatial distribution maps (SDMs) obtained from APT data. SDMs are statistical analyses of the interatomic distances used to reveal partial crystallographic information within APT data (Methods)[35–37]. ML, especially deep learning, has been used across microscopy and microanalysis to speed up and improve the repeatability of analyses and reveal elusive details[38,39]. In APT, it has been used for crystallographic orientation identification and improving microstructural feature extraction[40–42]. SDMs have been applied to long-range-ordered precipitates[17,37,43–45] but not to analyse CSRO.

## Results

### Conventional APT analysis

We performed APT on body-centred-cubic (BCC) Fe-18Al alloys annealed at two temperatures for 2 weeks (Methods, Supplementary Fig. 1). Figure 1a is an example of a reconstructed APT analysis along the (002) zone axis. Figure 1b is a detector hit map showing the corresponding symmetries of low-index sets of atomic planes[46], which are highlighted in Fig. 1c. A $2 \times 2 \times 2\,nm^3$ voxel was examined through SDMs along the depth, i.e. z-SDMs (Methods), for all possible pairs of elements, as plotted in Fig. 1d. The peak-to-peak distance is the same for all pairs, and the reconstruction was calibrated[47,48] to have the interspacing of 0.144 nm as expected for this composition[49].

A dedicated search for CSRO has been applied to a similar Fe-18Al alloy, but no concrete information was reported[16,17]. Other conventional analyses of APT data to extract CSRO include isosurfaces[42,43], 1st, 2nd, and 3rd nearest neighbours[16,50], and frequency distribution analysis[13,31]. Results from the latter two are displayed in Fig. 1e–g, but they do not provide evidence for CSRO, which can be attributed to a high fraction of reconstructed atoms shifting beyond their 1st nearest neighbours after field evaporation and data reconstruction[34]. The lateral resolution of APT is hence too low to detect CSRO with approaches that probe the reconstructed data assuming an isotropic distribution of atomic species[31,35]. However, most atoms at low-index planes do not shift beyond their 1st nearest neighbours along the depth direction[34], leaving an unexplored opportunity to analyse CSRO, as already achieved for long-range orders utilizing ML[42,43]. Please refer to Supplementary Notes for a detailed discussion on the reconstruction quality of APT data for CSRO analysis.

### Workflow of machine learning enhanced APT

Figure 2a summarises the complete framework of our proposed ML-classifier to distinguish the BCC matrix and different CSRO structures using z-SDMs. A synthetic bank of z-SDMs (Methods) was built for the

three crystal structures expected from the Fe-Al binary phase diagram[15,51], namely BCC, D0₃, and B₂ (FeAl) as detailed in Fig. 2b–e. Each structure exhibits a specific pattern in the z-SDMs (Fig. 2e), which enables classification between BCC matrix, D0₃-CSRO and B₂-CSRO. Although the 3D structure of CSRO is not fully the same as the relevant ordered one, they have a similar 1D z-SDM signature along specific direction. So, these simulated patterns based on ordered structures can be applied to train a recognition model aiming at CSRO. A total of 10,000 representative data samples were generated with diverse data (Methods, Supplementary Table 1, Supplementary Notes). Each continuous z-SDM was discretised using 93 data points into the input layer.

Then, we trained and validated a one-dimensional CNN (Methods, Supplementary Fig. 2a) with a five-fold cross-validation procedure using 90% of the synthetic dataset. The remaining 10% was used to test the CNN (Test I). The uncertainty of the model was calculated based on the five models obtained by five-fold cross-validation. For comparison, we also trained a random forest using the same synthetic data and a five-fold cross-validation procedure to perform the same classification task. In parallel, we scanned the reconstructed APT data, and generated experimental z-SDMs for the Fe-Fe and Al-Al elemental pairs. Afterwards, the data were pre-processed with a pipeline of transformations including curve smoothing and background reduction (Methods). Two examples of the original and optimized experimental curves are presented in Supplementary Fig. 3.

Both algorithms exhibited almost 100% training, validation, and test (Test I) accuracies (Supplementary Fig. 2b–d). The two models were further tested using 148 $2 \times 2 \times 2\,nm^3$ representative experimental z-SDMs after pre-processing with the labels given manually (Test II). Their classifications were evaluated by the area under the curve (AUC) of the receiver operating characteristic curve (ROC) (Methods). The CNN exhibited high AUC values and low uncertainties, i.e., $0.95 \pm 0.01$, $0.95 \pm 0.01$, $0.97 \pm 0.01$ for BCC, D0₃-CSRO, B₂-CSRO (Supplementary Fig. 4a), respectively. The false positive and false negative rates are low for D0₃-CSRO and B₂-CSRO, suggesting a limited influence on the overall analysis. These scores suggest that the CNN is also able to successfully classify experimental data. In comparison, the random forest algorithm only led to low AUC and high uncertainties for each class (Supplementary Fig. 4b). Only CSRO based on CNN will be discussed in the following.

To understand where the CNN model is focusing, we applied gradient-weighted class activation mapping[42,52]. Its output is a heatmap for a given class label. As shown in Supplementary Fig. 5, the model mainly looks at the specific peaks of the z-SDMs that can be used to accurately classify the three classes, i.e., the peaks at the zones close to the ΔZ with ±0.144 and ±0.432 nm for BCC and D0₃-CSRO, and those close to the ΔZ with 0, ±0.288 or ±0.576 nm for the B₂-CSRO.

We then tested our ML-APT recognition model using large-scale Fe-Al APT artificial data as ground truth (Methods). D0₃ domains, with a minimum diameter of 0.7 nm, can be distinguished well from the BCC matrix with Pearson's correlation coefficients (PCC) above 0.8 when using $1 \times 1 \times 1\,nm^3$ voxel (1-nm³ cube) with a 0.5 nm stride (Supplementary Fig. 6). The morphologies of the identified D0₃ domains, as described by their aspect ratio and oblateness reported in Fig. 2f, appear close to spheres. Some are slightly stretched or compressed, likely because of the voxelization. Figure 2g is a histogram of CSRO domains vs. size, which essentially follows the simulated data with a PCC of 0.67. The calculated number density of the identified D0₃ domains is $5.71 \times 10^{25}\,m^{-3}$, which is within 11% of the actual value. Even in a truly random yet concentrated solid solution, some local environments similar to CSRO randomly form, with no specific ordering driving force. We also applied this ML-APT recognition model to a randomized dataset (Method, Fig. 2g), enabling the identification of randomly-occurring small CSRO domains. The PCC of 0.39, in this case, shows that our ML model prediction (PCC 0.67) can identify the

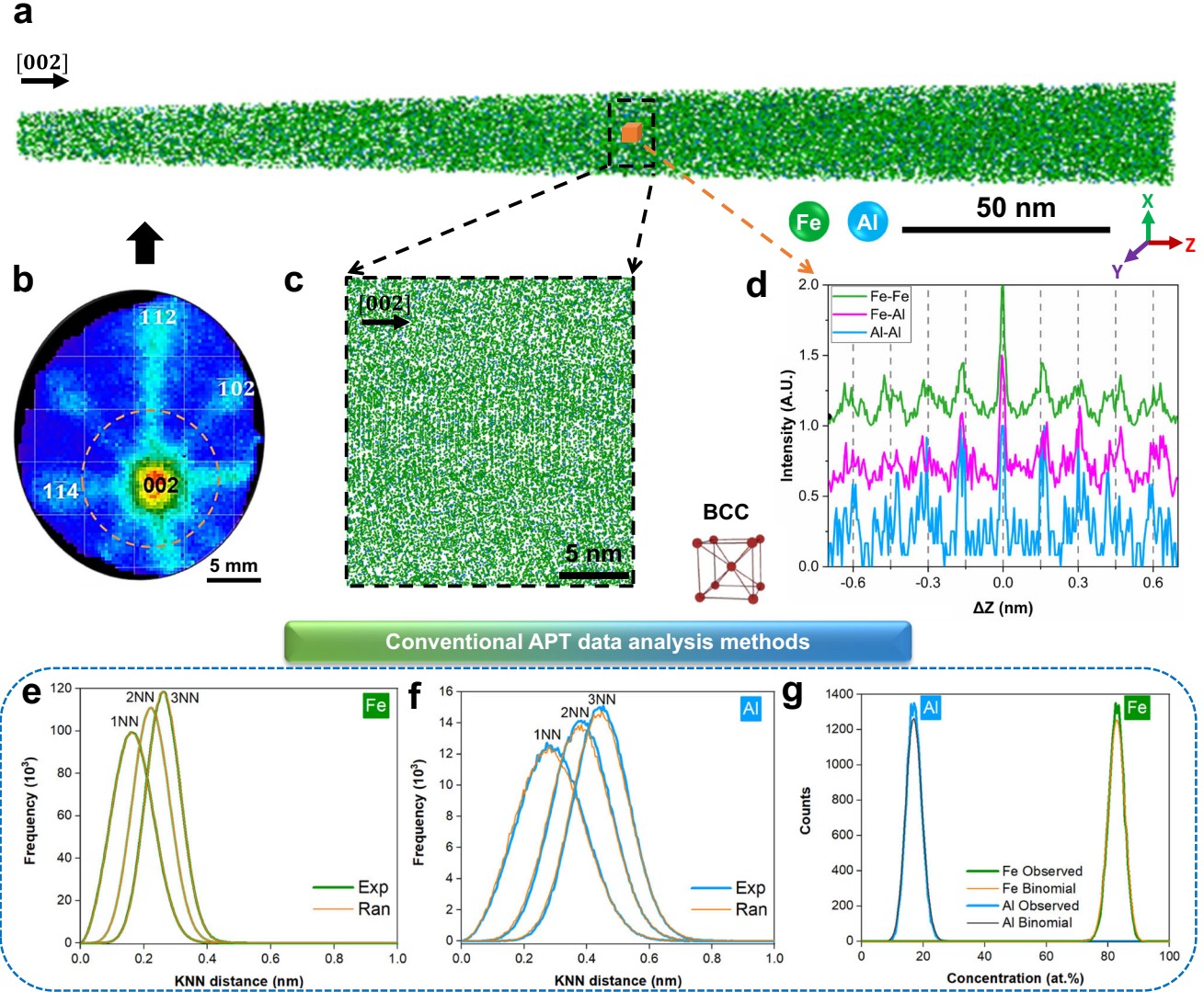

**Fig. 1 | APT data of Fe-18Al alloy annealing at 523 K for 14 days. a** APT reconstruction along the [002] zone axis. **b** two-dimensional detector hit map. Four crystallographic poles were indexed, corresponding to (002), ($\bar{1}$02), ($\bar{1}\bar{1}$2), (1$\bar{1}$4), respectively. **c** A close-up of a thin slice in **a** along [002]. **d** The z-SDMs of Fe-Fe, Fe-Al, Al-Al pairs in a $2 \times 2 \times 2$ nm$^3$ voxel in **a**. This signature corresponds to the BCC structure. Typical APT analysis methods: **e, f** K-nearest neighbour (KNN) distance analysis (k = 1, 2, and 3) of Fe-Fe and Al-Al atoms, respectively, and Exp and Ran refer to the results obtained from experimental and random-labelled datasets, respectively; **g** Frequency distribution analysis of Fe and Al atoms in comparison to the theoretical binomial random distribution.

underlying order and distinguish between CSRO and random structures. The same test procedure was applied to large-scale Fe-Al artificial APT data with $B_2$-type domains (Supplementary Fig. 7). Our ML-APT model distinguishes $B_2$ domains from the BCC matrix in terms of spatial distributions (PCC > 0.9), morphology, and size distributions (PCC = 0.62). Interestingly, only limited, small $B_2$ domains are detected in the randomized dataset, suggesting that it is more difficult to detect statistical $B_2$ domains than $D0_3$ ones.

**Application to experimental datasets**

Our ML-APT recognition model was subsequently applied to experimental datasets scanned with a 1-nm$^3$ cube and 0.5-nm stride. Here, we take the data shown in Fig. 1a as an example. In total, 653,944 experimental z-SDMs were generated. The distributions of the predicted $D0_3$-CSRO and $B_2$-CSRO probabilities corresponding to these experimental z-SDMs from $1 \times 1 \times 1$ nm$^3$ voxels and corresponding uncertainties are close to zero (Supplementary Fig. 8b, d). The probability for a certain CSRO is then estimated from the sum of the predicted probabilities of the 8 overlapped 1-nm$^3$ cubes since the stride was only 0.5 nm, and hence ranges between 0 and 8. The determined

classification thresholds are 3.75 and 4 for $D0_3$-CSRO and $B_2$-CSRO, respectively (Methods, Supplementary Fig. 9).

Figure 3a shows coloured regions individually identified with a $D0_3$-CSRO in the dataset shown in Fig. 1a, along with two close-ups viewed from two perpendicular directions. By using a cluster-finding approach and a best-fit ellipsoid (Methods), the aspect ratio of each CSRO domain is plotted against oblateness in Fig. 3b, evidencing an average near-spherical morphology. The statistical size distribution of $D0_3$-CSRO domains is given in Fig. 3c. Most of the $D0_3$-CSRO domains only contain less than 50 APT-counted atoms corresponding to approximately 1-nm$^3$ cube. Similarly, the obtained $B_2$-CSRO domains are also visualized (Fig. 3d) and their morphologies are also very close to spherical (Fig. 3e). Most of the $B_2$-CSRO domains also contain less than 50 APT-counted atoms (Fig. 3f). Note that most of the previous studies expected the existence of $D0_3$-CSRO rather than $B_2$-CSRO in this alloy[21,22].

We applied ML-APT to data obtained from Fe-18Al annealed for 14 days at either 873 K or 523 K (Methods). The results from the chemically-randomized datasets are compared. The statistical results are summarized in Fig. 4a and b, which compare the number density

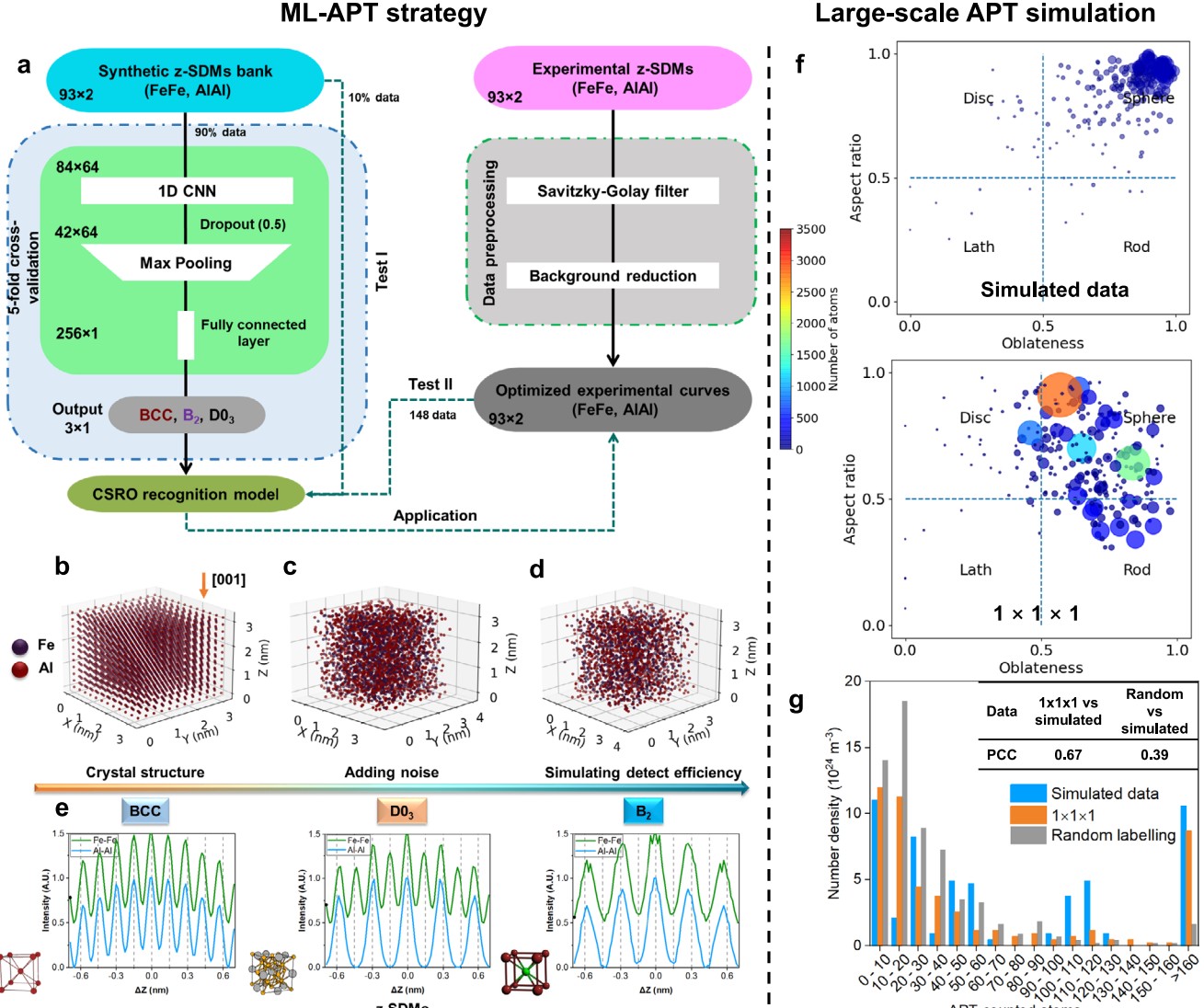

**Fig. 2 | Machine-learning-enhanced APT strategy to find CSRO and its application in large-scale APT simulation data. a** Flowchart of the proposed ML framework to find CSRO within APT data. Procedure to build three classes of crystal structures and generate relevant synthetic z-SDMs: **b** Generating supercell. **c** Atoms shift from theoretical sites in x, y, z directions. **d** Randomly discarding some of the atoms. **e** Examples of simulated z-SDMs of Fe-Fe and Al–Al pairs. Relevant crystal cells are enclosed. Test of the obtained ML-APT recognition model in large-scale Fe-Al APT simulation: **f** Morphology maps of the simulated CSRO domains and detected ones via the proposed model using $1 \times 1 \times 1$ nm³ scanning cubes. The size and colour of one circle denote the number of atoms within one domain. **g** Number densities versus APT-counted atoms corresponding to simulated and recognized CSRO domains. The result from the chemically-randomized dataset (Methods) is compared, and the Pearson's correlation coefficients (PCC) are listed in the inserted table.

distributions of the two types of CSRO domains with different sizes for these annealing conditions. Both states exhibit a high PCC (0.99) for the detected $D0_3$-CSRO compared to random, suggesting no substantial ordering beyond homogenous, statistically-random CSRO in this concentrated alloy. For $B_2$-CSRO, there is no obvious deviation from random at 873 K (PCC is 0.99), but a distinct difference appears at 523 K (PCC is 0.97). The number densities of the smaller (≤50 atoms or 1-nm cube) and larger (>50 atoms) domains increase by $8.74 \times 10^{23}$ and $4.18 \times 10^{23}$ m⁻³, respectively, as the annealing temperature decreases from 873 K to 523 K. The total number density of $B_2$-CSRO increases by $1.29 \times 10^{24}$ m⁻³. This suggests the formation and growth of non-statistical $B_2$-CSRO at 523 K. Due to its small size, these domains belong to CSROs not ordered ones.

We further verified our method in a Fe-19 at.% Ga alloy, in which previous TEM work[53] confirmed the existence of the tetragonally modified $D0_3$-CSRO. We collected the relevant APT data from the same material (Methods). The same model as for Fe-Al was applied and the

corresponding analyses are summarised in Supplementary Fig. 10, which confirms the existence of non-statistical modified $D0_3$-CSRO with a PCC of 0.977 not $B_2$-CSRO with a PCC of 0.993. The sizes of modified $D0_3$-CSRO are a little larger than those of $B_2$-CSRO in Fe-Al and some domains fall within the scale of medium-range order (2 to 3 nm). This proves that the ML-APT method can recognize $B_2$- and $D0_3$-CSRO and the finding of $B_2$-CSRO is reliable in the Fe-Al alloy.

To validate that the alloys we investigated exhibit CSRO, i.e. the K-state in these Fe-18Al alloys[14,19], we measured the nano-hardness and resistivity changes, shown in Fig. 4c. When the annealing temperature is decreased from 873 K to 523 K, the nano-hardness decreases from $3.05 \pm 0.15$ GPa to $2.81 \pm 0.21$ GPa at the {002} grain and from $2.75 \pm 0.05$ GPa to $2.70 \pm 0.04$ GPa at the {011} grain, both of which are equivalent within standard deviations. Based on the classical strengthening model (Methods) using the size distribution of CSRO domains (Fig. 4a, b), we estimated their contributions to be approximately 17.34 MPa and 21.08 MPa, at 873 K and 523 K, respectively,

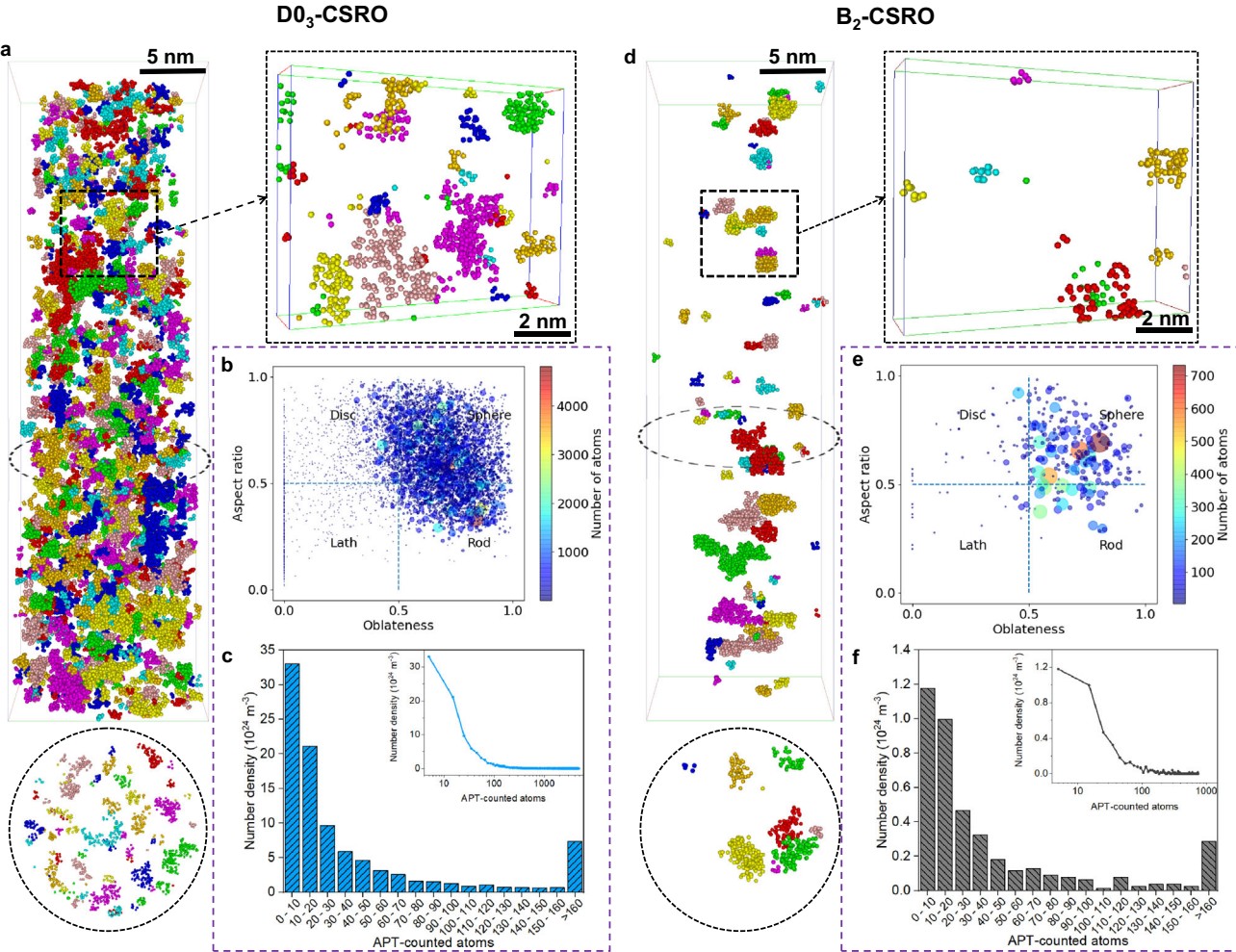

**Fig. 3 | Two types of CSRO in one Fe-18Al alloy annealing at 523 K for 14 days.** **a**, **d** Part of $D0_3$-CSRO and $B_2$-CSRO tomography from the same dataset, respectively. Different CSRO domains are marked using different colours. For each tomography, two thin slices of thickness 2 nm are highlighted. **b**, **e** Distributions of morphologies of $D0_3$-CSRO and $B_2$-CSRO domains, respectively. The size and colour of one circle denote the number of atoms within one domain. **c**, **f** Histograms of the size distributions of the identified $D0_3$-CSRO and $B_2$-CSRO domains, respectively. Another format with the logarithmic-scale x-axis is provided in the inserted image.

matching with experimental observations considering the experimental uncertainty. The resistivity rises by 5.3%, and previous reports attributed this anomalous resistivity change (~4% increase at 553 K for 1 day) only to $D0_3$-CSRO[16,21,54]. Our work proves that this resistivity change lies in the formation and growth of non-statistical $B_2$-CSRO $(1.29 \times 10^{24}\ m^{-3})$ at 523 K in this specimen.

## Discussion

The discovered $B_2$-CSRO is reliable considering the change in the electrical resistivity (Fig. 4c), the large-scale APT simulations with $B_2$-type domains (Supplementary Fig. 7), and the statistically significant difference between the observation and random state (PCC = 0.97 in Fig. 4b). In fact, previous X-ray scattering experiments and phase diagram calculations[20,55] indicated the existence of a narrow $B_2$ zone between BCC and $D0_3$, which directly supports our finding. It was thought that the $B_2$-FeAl structure, having aluminium atoms at second nearest neighbour distances, is an intermediary between disorder and the $D0_3$-Fe$_3$Al structure, where aluminium atoms are at third nearest-neighbour distances[20]. One would expect the formation of the FeAl as a transient in the formation process of the Fe$_3$Al superlattice, which is finally supported directly by our experimental evidence. The recognition ability on $D0_3$-CSRO of the proposed method is also verified in chosen Fe-Ga alloy as a ground truth (Supplementary Fig. 10).

Although TEM-based methods have been attempted to characterise CSRO, there are several limitations worth considering. First, the TEM community has suspected that the observed electron reflections could originate from other factors, including thin film effects, surface steps, surface oxides, and planar defects, rather than CSRO[56,57]. We performed aberration-corrected scanning transmission electron microscopy (STEM) experiments (Methods) to confirm the existence of CSRO in Fe-Al from the [110] zone axis, but the obtained fast Fourier transform patterns mostly match well with surface polycrystalline oxide films that inevitably exist on TEM specimens, not $D0_3$- or $B_2$- CSRO (Supplementary Fig. 11). Moreover, the TEM-based method cannot provide the quantitative distribution of CSRO in metallic materials, which varies with the specimen thickness[57]. Hence, quantifying CSRO configurations, sizes, and morphologies requires three-dimensional analytical imaging, which pushed us to develop the current method to enable quantitative assessment of CSRO from APT in a statistically-relevant way.

The performance of ML-APT is mainly limited by the APT data quality, especially its spatial resolutions. ML-APT will hence face limitations in the accurate detection of CSRO domains with fewer than 20 APT-reconstructed atoms. In the future, modelling the atom evaporation process[58,59] could improve the data quality (including detection efficiency and spatial resolutions) to maybe allow for more accurate recognition of CSRO using or extending the proposed ML-

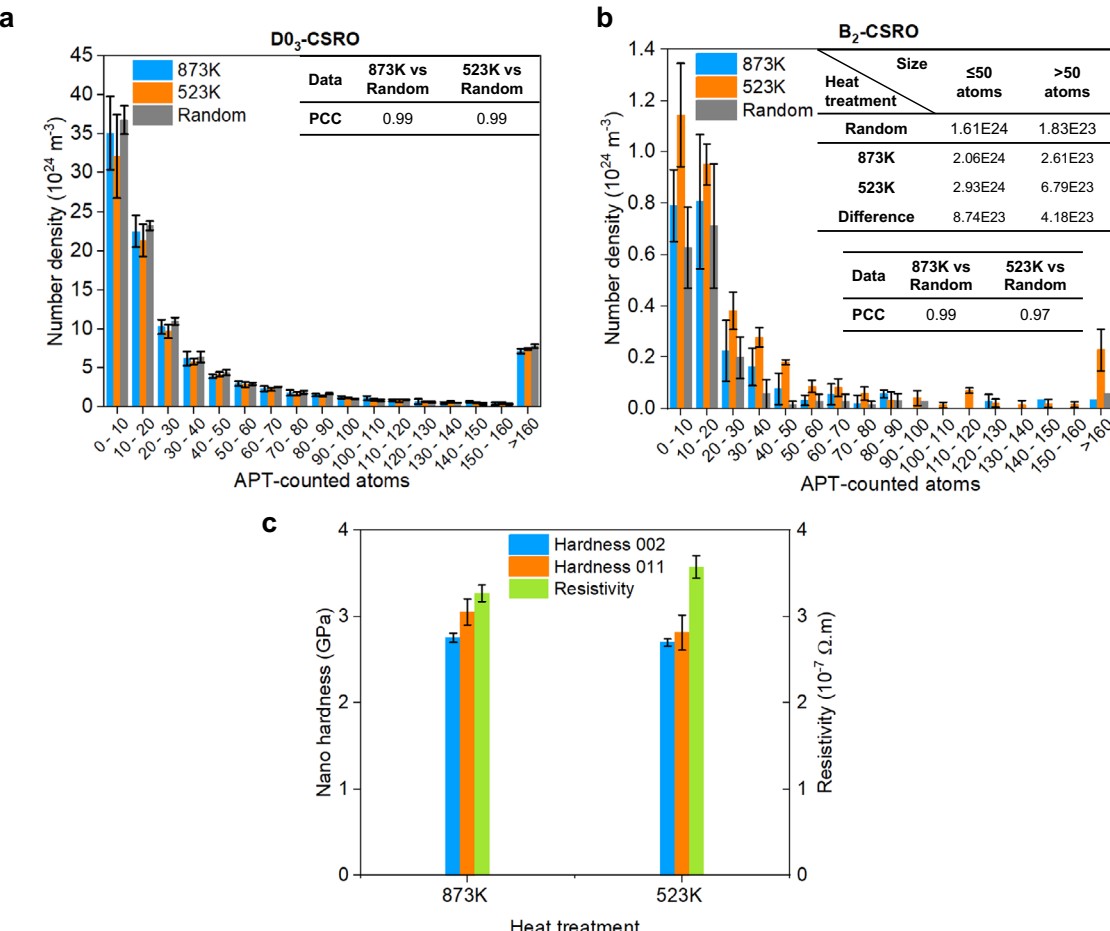

**Fig. 4 | Quantitative CSRO evolutions and property changes of Fe-18Al alloys with different heat treatments. a**, **b** Number density distributions of different sizes of $DO_3$-CSRO and $B_2$-CSRO, respectively, for two annealing temperatures for 14 days. The results from the chemically-randomized dataset are compared with the PCC values. The detailed changes in number densities of smaller and larger $B_2$- CSRO are listed in the inserted table. The value of 50, ideally corresponding to 1-nm$^3$ cube, is set as the dividing point. **c** Changes of nano-hardness and electrical resistivity at different annealing temperatures for 14 days. The values of nano-hardness at the {002} and {011} grains are given. All error bars (standard deviations) were obtained by at least three measurements.

APT method, as we could include in the training additional possible distortions of the features associated to CSRO. Furthermore, the present ML-APT approach requires prior knowledge of possible CSRO structures in the studied systems. Were this method to be applied to more complex alloy systems, like medium/high entropy alloys, the limited or absent prior knowledge of CSRO configuration could preclude identification and quantification.

This current finding addresses a long-standing question in Fe-18Al alloy and attributes the property changes upon heat treatment to the occurrence of non-statistical $B_2$-CSRO instead of the expected $DO_3$-CSRO, which has not been found by TEM-based methods. With the breakthrough in imaging CSRO in 3D by ML-APT, this approach can be extended to not only solve K-state phenomena in other alloying systems as mentioned above but also reveal CSRO in compositionally-complex alloys. These will help integrate CSRO into the design of future high-performance materials.

In summary, we managed to overcome the current limitation in imaging CSRO by APT by using ML. Our ML-APT strategy makes full use of the highest-quality, near-atomic-resolution of APT, combined with its high elemental analytical ability, which is unattainable by other techniques. This approach precisely reveals the morphology and size distributions of multiple types of (non-)statistical CSRO, in 3D and with near-atomic resolution, thereby enabling us to rationalise their origin and microstructural influence. We showcase this strategy on the open question in the context of the K-state Fe-Al system by establishing

relationships between the compositions, processing, CSRO and properties. In this case, the functional property is more sensitive to the formation of a limited number of non-statistical $B_2$-CSRO as compared to the mechanical property. However, whether abundant CSRO can affect the mechanical properties is still controversial, especially in compositionally-complex alloys[12,60] and we are now poised to characterise the CSRO in such complex alloy systems, and exploit short/medium/long-range ordering phenomena in newly designed advanced metallic materials.

## Methods
### Sample preparation
An alloy with a nominal composition of Fe-18Al (at.%) was arc-melted from pieces of raw metals with high purity (>99.8 wt.%) in a water-chilled copper crucible under an argon atmosphere. Then, the as-received sample was cut into pieces and annealed at 873 K and 523 K for 14 days, respectively, under argon atmosphere, followed by water quenching. The annealed samples were ground and polished for microstructure analysis as well as mechanical and functional properties tests. The backscattered electron images confirm that there are no obvious precipitates at both samples.

### APT experiments
For the Fe-Al, electron backscatter diffraction was first applied to determine the desired grain site along the [002] zone axis. Then, the

needle-like specimens for APT along [002] were prepared using a FEI Helios focused ion beam (FIB) with a Xenon plasma ion source. Note that using a FIB with the more commonly-utilized Gallium source was excluded because of the potential Gallium pollution in the Fe-Al alloy. The APT measurements were performed on a CAMECA Inc. LEAP 5000XR with a 52% detection efficiency. The APT experiments were carried out in voltage pulsing mode at 40–60 K, a detection rate of 0.2%, a pulsing fraction of 20%, and a pulsing rate of 200 kHz. Note that a temperature range from 80 K to 30 K was tried to increase the electric field to increase probability of the occurrence of $Al^{2+}$ and $Al^{3+}$ to avoid the peak overlap between $^{54}Fe^{2+}$ and $Al^+$ at 27 Da according to the bulk composition analysis (Supplementary Table 2)[16,17]. For each state, three APT datasets were collected for statistical analysis. AP suite 6.1 was used to make initial reconstruction and visualization. Two important reconstruction parameters, i.e., the field factor and image compression factor, were calibrated using the method suggested by Gault et al.[47,48]. The effect of H, C, and O in Supplementary Table 2 on the recognition of CSRO can be ignored.

For the Fe-Ga, the needle-like APT specimens were obtained from the single-crystal sample[53] along [002] using the same plasma FIB. The APT measurements were performed on a CAMECA Inc. LEAP 5000XS with a 80% detection efficiency in laser pulsing mode at 1.0% detection rate, 30pJ laser energy, and 200 kHz pulse rate. Three APT datasets were collected for statistical analysis. Other data analysis are same as the Fe-Al.

## Spatial distribution maps

SDMs are produced by establishing the vectors between each atom and its surrounding neighbours in 3D space[35–37]. These vectors are then accumulated into two-dimensional (2D) histograms (projected onto a plane, commonly the xy, or xz plane), or one-dimensional (1D) histograms (along the z-axis). Here, the 1D z-SDMs (see Fig. 1d) were employed to focus on the depth-direction structural information.

## Synthetic z-SDMs bank

Various possible configurations were simulated, around BCC, $DO_3$, and $B_2$ (Fig. 2b–e) using the following approach. First, a supercell with $3 \times 3 \times 3$ nm$^3$ was generated (the lattice constant was set as 0.287 nm[49]). Second, the atoms were shifted in x, y, z reconstruction directions according to Gaussian distributions to simulate the anisotropic spatial resolutions (depth resolution is better than the lateral direction). Third, a certain fraction of atoms was randomly discarded to simulate the imperfect detection efficiency met in APT data. Finally, 5000 z-SDMs of three structures were generated. The differences in the peak-to-peak distances of the Fe-Fe and Al-Al pairs allow one to make classification (Fig. 2e). The peak positions of the experimental z-SDMs were not always at their theoretical sites. Thus, we augmented the dataset by adding additional 5000 synthetic samples in which the peak-to-peak distance was randomly modified by ±0.06 nm. All parameters for generating simulated z-SDMs are listed in Supplementary Table 1.

## Experimental data pre-processing

The data pre-processing procedure includes using the Savitzky-Golay filter[61] to smooth the original z-SDM curve and make a background reduction. For the Savitzky-Golay filter, the length of the filter window and the order of the polynomial are set as 15 and 8, respectively, obtained by a tuning procedure in which different values are set and their performances on real data are evaluated. The background curve was calculated using[36]

$$y_p(i) = \min\left\{y_{p-1}(i), \frac{y_{p-1}(i+1) + y_{p-1}(i-1)}{2}\right\} \quad (1)$$

where $y_{p=0}$ is the smoothed z-SDM curve, and $y_p(i)$ is the value of the $i$th discrete point of the z-SDM at the $p$th iteration. The iteration is stopped when the first local minimum value of the smoothed z-SDM

(after x-axis is above 0) is lower than the maximum of $y_p(i)$ within the shade zone (Supplementary Fig. 3). The background as defined above was subtracted from the smoothed z-SDM curve to obtain the optimized z-SDM.

## CSRO recognition model based on 1D CNN and random forest algorithms

As detailed in Supplementary Fig. 2a, the 1D CNN[62,63] consists of a convolutional layer with filter number, kernel size and stride of 64, 10 and 1, respectively, followed by a dropout layer with the rate of 0.5 used to avoid overfitting. Then, a max-pooling layer[62] with a pool size of 2 is added to highlight the most important features. A fully-connected layer with 256 neurons is employed to generate an output corresponding to the three crystal structures. All layers used the activation function of ReLu, except the output layer which used softmax for classification purposes. The categorical cross-entropy was chosen as the loss function, which is commonly used to train a multi-classifier. We used a batch size of 32 and an optimized learning rate of 0.001 during training, and the Adam optimizer was used to minimize the loss function. Note that the reported hyperparameters' values in the 1D CNN are the result of a thorough tuning procedure based on the training, validation, and test results on synthetic and real datasets (like the tested convolutional layer number: 1 - 5; filter number: 8 - 64; kernel size: 3–20, neurons of full layer: 64 - 1000; learning rate: 0.001 - 0.1). For the random forest classifier[64], the number of trees in the forest was set to 200 based on the training, validation, and test results on synthetic and real datasets (the tested number of trees: 100 - 300). The CNN was implemented using Keras 2.2.4 with the TensorFlow 1.13.1 backend on Python 3.7. The random forest was performed using scikit-learn 0.23.2 using the default parameters except for the number of trees.

## Evaluation of classification ability based on ROC analysis

The ROC analysis[65,66] was applied to evaluate the performance of the multi-label classifier. In Supplementary Fig. 4, one curve is plotted based on two critical evaluation parameters: true positive rate and false positive rate. The top left corner of this curve is an ideal point, i.e., the true positive rate and false positive rate are 1 and 0, respectively. For each label, we plot its ROC curve regarding each element of the label indicator matrix as a binary prediction. Five models from five-fold cross-validation provide five ROC curves for each class, and the mean value and standard deviation are plotted. The mean AUC and its standard deviation of each class are also provided. Note that the ROC is insensitive to the distribution of data, making it suitable for this work[67].

## Large-scale Fe-Al APT simulation

We built a perfect BCC supercell ($10 \times 10 \times 50$ nm$^3$) with $DO_3$-type or $B_2$-type domains (the diameters vary from 0.7 to 2.0 nm). Then, the atoms were shifted in lateral and depth reconstruction directions according to Gaussian distributions. According to the previous APT evaporation simulation in alloys and concentrated solid solutions[34], the maximum atoms shift in the lateral and depth directions were set as 5th and 1st nearest neighbour distances, respectively. The detection efficiency was set to 52%. Finally, the simulated $DO_3$-type and $B_2$-type domains are shown in Supplementary Figs. 6a and 7a, respectively. The ML-APT recognition model obtained was applied to the simulated APT data using different scanning strategies (Supplementary Figs. 6b, c and 7b). The 3D recognition results were compared via the 2D distribution of the number of atoms along Z and Y directions (Supplementary Figs. 6d, e and 7c, d), morphology and size distributions (Fig. 2f, g and Supplementary Fig. 7e, f).

To figure out the impact of the randomly-formed CSRO from the truly random solid solution, we randomly swapped the elemental identities of data points while retaining the original x, y, and z coordinates. Then, the same ML-APT recognition model was applied to the random dataset to obtain the randomly-formed CSRO domains. This is

compared with the results above using Pearson's correlation coefficient (PCC). PCC is a statistic that evaluates the linear correlation between variables X and Y, given by[68]:

$$PCC = \frac{\sum_{i=1}^{N}(x_i - \widetilde{x})(y_i - \widetilde{y})}{\sqrt{\sum_{i=1}^{N}(x_i - \widetilde{x})^2}\sqrt{\sum_{i=1}^{N}(y_i - \widetilde{y})^2}} \quad (2)$$

where N is the data size. $x_i$, $y_i$ are the individual data points. $\widetilde{x}$, $\widetilde{y}$ are the mean values. PCC varies from −1 to 1, suggesting the linear correlation extent of the two variables. 1 represents a complete positive linear correlation, 0 is no linear correlation, and −1 is a complete negative linear correlation. Note that the comparison using a chemically-randomized dataset is also applied for the experimental dataset, as shown in Fig. 4.

## Classification thresholds

Supplementary Fig. 9a, b presents the frequency distributions of two kinds of CSRO structure probabilities of the 0.5-$nm^3$ voxels (named $P_{0.5}$). The data from three zones (1, 2, 3) in Supplementary Fig. 9a were analysed and the corresponding z-SDMs are shown in Supplementary Fig. 9c. For the z-SDMs of Fe-Fe pairs, the peaks are unclear, which also occurs in the large-scale APT simulation data. This is attributed to the imperfect lateral and depth resolutions. For the z-SDMs of Al-Al pairs, there is an obvious trend to be close to the signature of $DO_3$-CSRO except zone 1. Thus, the separating line between zones 1 and 2 was regarded as the threshold between $DO_3$ and others, i.e., 3.75. Similarly, zones 1–3 in Supplementary Fig. 9b were also analysed and the corresponding z-SDMs are plotted at Supplementary Fig. 9d. The $B_2$ signature becomes clear after zone 1, and, thus, 4 was seen as the threshold. Finally, the z-SDMs of each kind of CSRO based on the chosen threshold exhibits a clear signature (zone 4), suggesting the excellent CSRO classification ability.

## Quantification of CSRO domains

We made a quantitative analysis of the identified CSRO domains based on the cluster analysis algorithm included in AP suite 6.1. The approach consists of choosing a maximum separation distance between the clustered elements ($d_{max}$) and a minimum number of ions in the cluster ($N_{min}$)[50,69]. After getting the optimized parameters in the simulated APT dataset (Fig. 2f, g) these values were further fine-tuned in the real APT dataset by visual observation. The determined values of $d_{max}$ and $N_{min}$ are 0.4 nm and 3 ions, respectively, which was applied to all real APT datasets. Then, the obtained count of CSRO domains was divided by the total volume to calculate the number density of CSRO, $N_v$.

To describe the shape of each CSRO domain, we first define its centre of mass in a best-fit coordinate system, $r_{com}$, by

$$r_{com}(x_{com}, y_{com}, z_{com}) = \frac{\sum_{i=1}^{N_c} m_i r_i(x_i, y_i, z_i)}{\sum_{i=1}^{N_c} m_i} \quad (3)$$

where $m_i$ and $r_i$ denote the mass and coordinates (in the best-fit coordinate system) of each atom in the domain and $N_c$ is the total number of atoms in the domain. Then, the three radii of gyration in the best-fit ellipsoid system, ($R_{g1} > R_{g2} > R_{g3}$), were calculated. The shape of the domain is expressed by the plot of the oblateness vs aspect ratio, defined by

$$Oblateness = \frac{R_{g3}}{R_{g2}} \quad (4)$$

$$Aspect\ ratio = \frac{R_{g2}}{R_{g1}} \quad (5)$$

## Nano-hardness and electrical resistivity measurements

The Nano-hardness tests were performed using the G200 nanoindenter at a strain rate of 0.1 $s^{-1}$ and an indentation depth of 1000 nm. The grains close to {002} and {011} were selected and the hardness values from 50 measurements within the specific grains were measured to calculate the statistical results for each sample. The electrical resistivity corresponding to each annealing temperature was measured on a Quantum Design Physical Property Measurement System with a temperature of 300 K and zero Tesla magnetic field. The resistivity values from 4 samples were obtained to calculate the statistical results for each state.

## Strength model

The main strength contribution can be divided into the intrinsic strength from the BCC matrix, solid solution strengthening, and precipitation strengthening from CSRO. The two formers are almost identical for the studied states. The precipitation strengthening from CSRO, $\sigma_p$, is estimated by the previously-developed precipitation strength model in terms of spherical precipitates[70–73]:

$$\sigma_p = 0.9M\mu b\sqrt{2\bar{r}N_V}\bar{f}^{\frac{3}{2}}\left(1 - \frac{1}{6}\bar{f}^5\right) \quad (6)$$

$$\bar{f} = \left(\frac{1}{r_c^{2k}}\int_0^{rc} r^{2k+1}\varphi_r d_r + \int_{rc}^{\infty} r\varphi_r d_r\right)\frac{1}{\bar{r}N_v} \quad (7)$$

where $M$ is the Taylor factor. $\mu$ is the shear modulus of $DO_3$ or $B_2$, and $b$ is the Burgers vector. $N_V$ is the total number density of CSRO. $\bar{r}$ is the mean particle radius and $r_c$ is the critical radius that the precipitates transform from shearable to non-shearable. $k$ is an empirical model parameter. $\varphi_r$ is the statistical distribution of the number of particles per volume as a function of the particle radius, $r$. Here, $M$=3, $\mu$=69.8 ($DO_3$) or 113.4 ($B_2$) GPa[74], $b$=0.25 nm, $k$=1, and $r_c$ = 5 nm were used[75].

## STEM experiments

Electron backscatter diffraction was first applied to determine the desired grain with the aimed orientation. Then, the STEM specimen was prepared using a Thermofisher Scios 2 FIB with a Gallium source. The parameters used for the final milling were 2 kV and 27pA to minimize the possible contamination and damage to the specimen. The thickness of each TEM specimen was directly measured under the FIB window after the final polishing. The atomic resolution STEM imaging was performed on a FEI Themis Z (60–300 kV) scanning transmission electron microscopy with double aberration correctors. STEM images were captured at 300 kV with a convergence semi-angle of 23.6 mrad and a screen current of 50 pA.

## Data availability

All data that support the findings are involved in this paper. An annealed Fe-Al APT demo data is provided in https://doi.org/10.6084/m9.figshare.23989050.

## Code availability

The major codes are available at GitHub address https://github.com/a356617605[76].

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

## Acknowledgements

This work was primarily supported by the Max Planck research network on big-data-driven materials science (BiGmax) and the research fellowship provided by the Alexander von Humboldt Foundation. We also acknowledge the Open Foundation of State Key Laboratory of Powder Metallurgy at Central South University, Changsha, China. Z.W. was supported by Natural Science Foundation of Hunan Province (Grant No. 2022JJ30712). I.B. was supported by the U.S. Dept. of Energy, Office of Science, Basic Energy Sciences # DE-SC0018962. J. Rao and M. J. Duarte are acknowledged for their great help with nano-hardness tests. Q. Du from SINTEF is acknowledged for a fruitful discussion. Y. Wu is acknowledged for his help in providing the Fe-Ga alloy.

## Author contributions

Y.L., Z.W., and B.G. conceived the project. Y.L. was the lead experimental/data scientist of the study. Z.W., I.B., and Y.L. proposed the Fe-Al system and prepared the materials, samples and heat treatments. Y.L., Z.W., and X.Z. performed the PFIB/APT experiments with the help of L.S. and B.G. Y.L. analysed the APT data with the help of L.S. and B.G. Y.L. programmed the machine learning framework with the help of Y.W., A.M., M.R., S.B., and T.C. Y.L. conducted APT simulations and strength modelling. L.Ha. and Z.R. performed the electrical resistivity measurements. X.L. performed the STEM experiments. L. Hu., R.D., Y.G., H.L., and J.N. contributed to the explanations of the experimental results. Y.L., Z.W., and B.G. wrote the manuscript. All authors contributed to the discussion of the results and commented on the manuscript.

## Funding

## Competing interests

The authors declare no competing interests.
