## [Peer Review File · Nature Communications]

REVIEWER COMMENTS

Reviewer #1 (Remarks to the Author):

The manuscript addresses an important topic in material science, focusing on the characterization of chemical short-range order (CSRO) in Fe-Al alloys, which are promising materials for various applications. The proposed approach of using machine learning (ML), particularly a convolutional neural network (CNN), to analyze CSRO based on APT data is innovative and demonstrates the potential to overcome the limitations of conventional techniques. However, there are some major issues with the manuscript in its current form, please see below

- 1) Overall, the manuscript poses some challenge to follow, some sections need to be rewritten to improve the readability and clarity. It would be helpful to explicitly state the significance of understanding CSRO in Fe-Al alloys, such as how this knowledge could lead to improved material design or impact specific applications in engineering or industry.
- 2) The manuscript contains a significant amount of technical information about the synthetic bank, model training, and classification results. To improve clarity and readability, consider breaking down complex concepts into more digestible sections or paragraphs. Ensure that the flow of information is logical and easy to follow.
- 3) While the authors mention the high accuracy and successful classification of CSRO using CNN, it would be valuable to explicitly discuss the significance and implications of these findings. How does the ability to detect CSRO impact the field of material science, and how might it lead to advancements in material design and performance?
- 4) The heatmap generated by gradient-weighted class activation mapping was mentioned, however the actual heatmap itself is not provided. Consider including the heatmap in the figures section to illustrate where the CNN model is focusing during the classification. More detailed explanations of how the model identifies specific features in z-SDMs and how these features relate to CSRO classification would enhance the interpretability of the ML approach.
- 5) While the ML model exhibits high accuracy and AUC values, there is no detailed exploration of the model's uncertainties. The inclusion of uncertainty estimates for the classification results would provide a more comprehensive understanding of the reliability and robustness of the ML approach.

6) The authors mention using a total of 10,000 data samples for training and validation, but details about the diversity of these samples, as well as any potential biases in the dataset, are missing. Additionally, the study may benefit from using a larger and more diverse dataset to ensure the model's generalizability to various Fe-Al alloy compositions and microstructures.

7) Consider including a brief section discussing the limitations of the proposed approach and potential areas for future research. Address any challenges and uncertainties that might arise when applying ML models or more complex material systems or real world applications.

8) It is stated that the spatial resolution of APT prevents precise imaging of the atomic ordering process below 1 nm in radius. This limitation may hinder the detection and classification of very small CSRO domains, and it should be acknowledged in the manuscript.

9) Lastly, the authors focus on the high accuracy and AUC values achieved by the ML model. However, it is essential to discuss potential cases of false positive and false negatives in the classification results and the impact they may have on the overall analysis.

Reviewer #2 (Remarks to the Author):

Dear Authors,

The manuscript under consideration is interesting in that it presents an investigation into an important yet challenging problem of identifying Chemical Short-Range Order (CSRO) in APT data. The utilization of machine learning (ML)-based techniques in the context of material characterization, analysis, and mapping is indeed a move in the right direction as it has the potential to shed new light on microstructural features that would be elusive through conventional means. Traditionally, characterizing CSRO in disordered crystalline matrices has posed formidable challenges due to the limitations of conventional techniques and ML-enhanced approaches has the potential to break these inherent resolution limits of APT and delve into the intricate sub-nanoscale features, which would address some of the issues unanswered before for instance, anomalous property changes during heat treatment of disordered crystalline alloys.

One interesting aspect of this work is the successful evidence of non-statistical B_2 -CSRO (FeAl) instead of the anticipated $D0_3$ -CSRO (Fe_3Al) in the $Fe_{18}Al$ alloy which demonstrates the method's efficacy in uncovering unexpected short-range ordering phenomena. The application of the proposed strategy to modified $D0_3$ -CSRO detected in the Fe-Ga system, also corroborated by other microscopy techniques, further strengthens the generalizability of the approach.

Having said that, I have a number of comments to make and a few questions expected to be answered which are as follows:

- Given that the ML model was trained on synthetic z-SDM dataset, how can we be confident that the model also works reliably for the experimental dataset, especially when we know that the field evaporation can be noisy (and, *to some minimal extent*, random) and hence the reconstruction procedure might not faithfully reproduce the exact same features found on the lost specimen? Couldn't you potentially be mistaking some reconstruction artefacts for CSRO? In short, can we trust the reconstruction algorithm to reproduce the original features or the discrepancies between the original and reconstructed does not considerably affect the results in the given problem of identifying CSRO?

- In the paragraph describing CSRO recognition model, it would be a good idea to describe input layer (number of neurons) and nature of the data being input (linking z-SDMs to the input layer)

- It would be advisable to go through a couple of research papers I have linked below.

<https://doi.org/10.1016/j.pmatsci.2021.100854>

<https://doi.org/10.1016/j.pmatsci.2021.100854>

Short-range ordering is also a matter of significant interest in HEAs and I think it would be a good idea to have a cursory reference to these as well.

- Line 227 "believable" might not be the right word. Line 271, is it ground truth?

- Line 174: What is LCO an abbreviation of?

Thank you and best regards

Response to Reviewers' Comments

Manuscript NCOMMS-23-25153

We would like to start by thanking the editor and reviewers for the valuable suggestions and comments. We have performed more work that has now been blended into the manuscript and which helps settle all concerns from the two reviewers, thereby strengthening our work.

Our response is structured as follows: The comments from the reviewers are copied below (in black), and for each comment, we provide a response and the corresponding manuscript modifications (in blue).

In the revised manuscript, more details, explanations, and discussions of our approach and findings were supplemented. Two supplementary Notes were added and two figures were updated with refined data analyses (Supplementary Figs. 4 and 5), as well as one figure was added (Supplementary Fig. 12). We finally pointed out the reliability and future development of our method.

The main revisions include:

- (1) Explanations of the uncertainty of the model (Comment (5) from the 1st referee);
 - (2) Details of the used dataset (Comment (6) from the 1st referee);
 - (3) Discussion on the effects of false positives and negatives on the classification results (Comment (9) from the 1st referee);
 - (4) Assessment of the reliability of the real data and model (Comment (9) from the 1st referee and Comment (1) from the 2nd referee)
-

Referees' comments:

Referee #1 (Remarks to the Author):

The manuscript addresses an important topic in material science, focusing on the characterization of chemical short-range order (CSRO) in Fe-Al alloys, which are promising materials for various applications. The proposed approach of using machine learning (ML), particularly a convolutional neural network (CNN), to analyze CSRO based on APT data is innovative and demonstrates the potential to overcome the limitations of conventional techniques. However, there are some major issues with the manuscript in its current form, please see below.

Response: Many thanks for your suggestive comments. We have adopted your suggestions and addressed all your concerns in the following part.

(1) Overall, the manuscript poses some challenge to follow, some sections need to be rewritten to improve the readability and clarity. It would be helpful to explicitly state the significance of understanding CSRO in Fe-Al alloys, such as how this knowledge could lead to improved material design or impact specific applications in engineering or industry.

Response: Many thanks.

For the readability and clarity, we have added subheadings to make it easier to follow for general readers (see below). Complex paragraphs have been divided into more digestible sections according to your next comment.

For the significance of understanding CSRO in Fe-Al and other alloys, as our response to your comment (3), we added: this current finding addresses a long-standing question in Fe-18Al alloy and attributes the property changes upon heat treatment to the occurrence of non-statistical B₂-CSRO instead of the expected D0₃-CSRO, which has not been found by TEM-based methods. With the breakthrough in imaging CSRO in 3D by ML-APT, this approach can be extended to not only solve K-state phenomena in other alloying systems as mentioned above but also reveal CSRO in compositionally-complex alloys. These will help integrate CSRO into the design of future high-performance materials.

Modifications:

(1) Headings and subheadings have been added:

- Introduction
- Results
 - Conventional APT analysis
 - Workflow of machine learning enhanced APT
 - Applications in experimental datasets
- Discussion

(2) The revision of complex paragraphs can be referred to the response to your next comment.

(3) We have added the significance of understanding CSRO in alloys (Page 14).

This current finding addresses a long-standing question in Fe-18Al alloy and attributes the property changes upon heat treatment to the occurrence of non-statistical B₂-CSRO instead of the expected D0₃-CSRO, which has not been found by TEM-based methods. With the breakthrough in imaging CSRO in 3D by ML-APT, this approach can be extended to not only solve K-state phenomena in other alloying systems as mentioned above but also reveal CSRO in compositionally-complex alloys. These will help integrate CSRO into the design of future high-performance materials.

(2) The manuscript contains a significant amount of technical information about the synthetic bank, model training, and classification results. To improve clarity and readability, consider breaking down complex concepts into more digestible sections or paragraphs. Ensure that the flow of information is logical and easy to follow.

Response: Many thanks for the suggestions. We have divided synthetic bank, model training, and classification results into different paragraphs (each section has a single paragraph in Pages 6-7). More descriptions and explanations of models have been added, such as the visualization of the obtained CNN model (reply to your comment (4)), the uncertainty of the model (reply to your comment (5)), the generalizability of our ML-APT model and the diversity of simulated data (reply to your comment (6)), and ROC curves (reply to your comment (9)).

Modifications: We have modified the technical information to make it logical and easy to follow (Pages 6-7). More descriptions and explanations of models can be referred to in the response to the following comments.

Pages 6-7: Fig. 2a summarises the complete framework of our proposed ML-classifier to distinguish the BCC matrix and different CSRO structures using z-SDMs. A synthetic bank of z-SDMs (Methods) was built for the three crystal structures expected from the Fe-Al binary phase diagram^{15,51}, namely BCC, D0₃, and B₂ (FeAl) as detailed in Fig. 2b-e. Each structure exhibits a specific pattern in the z-SDMs (Fig. 2e), which enables classification between BCC matrix, D0₃-CSRO and B₂-CSRO. Although the 3D structure of CSRO is not fully the same as the relevant ordered one, they have a similar 1D z-SDM signature along specific direction. So, these simulated patterns based on ordered structures can be applied to train a recognition model aiming at CSRO. A total of 10,000 representative data samples were generated with diverse data (Methods, Supplementary Table 1, Supplementary Notes). Each continuous z-SDM was discretised using 93 data points into the input layer.

Then, we trained and validated a one-dimensional CNN (Methods, Supplementary Fig. 2a) with a five-fold cross-validation procedure using 90% of the synthetic dataset. The remaining 10% was used to test the CNN (Test I). The uncertainty of the model was calculated based on the five models obtained by five-fold cross-validation. For comparison, we also trained a random forest using the same synthetic data and a five-fold cross-validation procedure to perform the same classification task. In parallel, we scanned the reconstructed APT data, and generated experimental z-SDMs for the Fe-Fe and Al-Al elemental pairs. Afterwards, the data were pre-processed with a pipeline of transformations including curve smoothing and background reduction (Methods).

Two examples of the original and optimized experimental curves are presented in Supplementary Fig. 3.

Both algorithms exhibited almost 100% training, validation, and test (Test I) accuracies (Supplementary Fig. 2b-d). The two models were further tested using 148 $2 \times 2 \times 2$ nm³ representative experimental z-SDMs after pre-processing with the labels given manually (Test II). Their classifications were evaluated by the area under the curve (AUC) of the receiver operating characteristic curve (ROC) (Methods). The CNN exhibited high AUC values and low uncertainties, i.e., 0.95 ± 0.01 , 0.95 ± 0.01 , 0.97 ± 0.01 for BCC, D0₃-CSRO, B₂-CSRO (Supplementary Fig. 4a), respectively. The false positive and false negative rates are also quite low for D0₃-CSRO and B₂-CSRO, suggesting their limited influence on the overall analysis. These scores suggest that the CNN is also able to successfully classify experimental data. In comparison, the random forest algorithm only led to low AUC and high uncertainties for each class (Supplementary Fig. 4b). Only CSRO based on CNN will be discussed in the following.

To understand where the CNN model is focusing, we applied gradient-weighted class activation mapping^{42,52}. Its output is a heatmap for a given class label. As shown in Supplementary Fig. 5, the model mainly looks at the specific peaks of the z-SDMs that can be used to accurately classify the three classes, i.e., the peaks at the zones close to the ΔZ with ± 0.144 and ± 0.432 nm for BCC and D0₃-CSRO, and those close to the ΔZ with 0, ± 0.288 or ± 0.576 nm for the B₂-CSRO.

(3) While the authors mention the high accuracy and successful classification of CSRO using CNN, it would be valuable to explicitly discuss the significance and implications of these findings. How does the ability to detect CSRO impact the field of material science, and how might it lead to advancements in material design and performance?

Response: Many thanks. We have added the implications of our work on other K-state phenomena in other alloys besides the Fe-Al. The CSRO would be beneficial to design future high-performance materials.

Modifications: We added a paragraph to discuss it.

Page 14: This current finding addresses a long-standing question in Fe-18Al alloy and attributes the property changes upon heat treatment to the occurrence of non-statistical B₂-CSRO instead of the expected D0₃-CSRO, which has not been found by TEM-based methods. With the breakthrough in imaging CSRO in 3D by ML-APT, this approach can be extended to not only solve K-state phenomena in other alloying systems as mentioned above but also reveal CSRO in compositionally-complex alloys. These will help integrate CSRO into the design of future high-performance materials.

(4) The heatmap generated by gradient-weighted class activation mapping was mentioned, however the actual heatmap itself is not provided. Consider including the heatmap in the figures section to illustrate where the CNN model is focusing during the classification. More detailed explanations of how the model identifies specific features in z-SDMs and how these features relate to CSRO classification would enhance the interpretability of the ML approach.

Response: Many thanks for the suggestion. We have added the heatmap itself using plasma colour map, as shown below. The heatmap corresponding to gradient-weighted class activation mapping is plotted using a red colour map. The more red, the more attention. The model mainly looks at the specific peaks of the z-SDMs that can be used to accurately classify the three classes, i.e., the peaks at the zones close to the ΔZ with ± 0.144 and ± 0.432 nm for BCC and D0₃-CSRO, and those close to the ΔZ with 0, ± 0.288 or ± 0.576 nm for the B₂-CSRO.

Supplementary Fig. 5 Visualization of the obtained CNN model on three classes of z-SDMs via gradient-weighted class activation mapping. **a** BCC. **b** D0₃. **c** B₂. The heatmap of z-SDMs of Fe-Fe and Al-Al pairs is given using a plasma colour map. The heatmap corresponding to gradient-weighted class activation mapping is plotted using a red colour map. The higher level of red colour indicates more attention. For the two formers, the model is looking at whether there are peaks at the zones close to the ΔZ with ± 0.144 and ± 0.432 nm. For the B₂, the model focuses on the zones close to the ΔZ with 0, ± 0.288 or ± 0.576 nm.

Modifications: We modified Supplementary Fig. 5 as shown upper. A more detailed explanation is given in Page 7.

Page 7: To understand where the CNN model is focusing, we applied gradient-weighted class activation mapping^{42,52}. Its output is a heatmap for a given class label. As shown in Supplementary Fig. 5, the model mainly looks at the specific peaks of the z-SDMs that can be used to accurately classify the three classes, i.e., the peaks at the zones close to the ΔZ with ± 0.144 and ± 0.432 nm for BCC and D0₃-CSRO, and those close to the ΔZ with 0, ± 0.288 or ± 0.576 nm for the B₂-CSRO.

(5) While the ML model exhibits high accuracy and AUC values, there is no detailed exploration of the model's uncertainties. The inclusion of uncertainty estimates for the classification results would provide a more comprehensive understanding of the reliability and robustness of the ML approach.

Response: The model's uncertainties are quite important and, in fact, we have considered it in this original manuscript. To make it clearer, we further highlight it with more descriptions in the revised version, including how to calculate uncertainty, the evaluation of uncertainty via ROC curve, the distribution of uncertainty in real data, and the extensive application of the model in the Fe-Ga alloy.

Modifications:

We highlight Page 6: The **uncertainty** of the model was calculated based on the five models obtained by five-fold cross-validation.

Page 6: The CNN exhibited high AUC values and low **uncertainties**, i.e., 0.95 ± 0.01 , 0.95 ± 0.01 , 0.97 ± 0.01 for BCC, D0₃-CSRO, B₂-CSRO (Supplementary Fig. 4a), respectively.

Page 9: The distributions of the predicted D0₃-CSRO and B₂-CSRO probabilities corresponding to these experimental z-SDMs from $1 \times 1 \times 1$ nm³ voxels and corresponding **uncertainties** are close to zero (Supplementary Fig. 8b, d).

For the exploration of the model, the same model was applied to the Fe-Ga system to make predictions and the results match well with previous TEM work (Page 10).

(6) The authors mention using a total of 10,000 data samples for training and validation, but details about the diversity of these samples, as well as any potential biases in the dataset, are missing. Additionally, the study may benefit from using a larger and more diverse dataset to ensure the model's generalizability to various Fe-Al alloy compositions and microstructures.

Response: We would like to answer your comment in terms of the range of the parameters for simulated data, the visualisation of the distribution of simulated data, and the application of the recognition model of Fe-Al alloys on Fe-Ga alloys.

First, as listed in Supplementary Table 1, the main parameters for simulating specific structures under different APT conditions are varied across a wide range of parameters. For the lateral resolutions (σ_x and σ_y), the defined range has included the most-encountered situation in APT data (see Fig. R 1). It is similar for the depth resolution (σ_z). Worse resolutions, i.e., leading to a loss of the capability to image atomic planes, were not considered because generating featureless patterns would be meaningless for training the ML model. Our ML model generally classifies these noisy and featureless

patterns as the BCC matrix to avoid bias from low-quality z-direction signals (see Supplementary Fig. 9). The simulated range of detection efficiency contains values for commercial APT apparatuses (0.35, 0.52, and 0.8) and includes worse conditions (e.g., 0.2). We even considered the effect of local peak shifts (-0.06~0.06nm) in the simulated SDMs database, which could reflect slight distortions arising from, e.g., the evolution of the projection parameters in the reconstruction model (Ultramicroscopy 132, 107-113 (2013); Ultramicroscopy 111, 1619-1624 (2011)). We tested the larger data size via either refining the original parameter range or extending this range slightly, but the performance of the model did not change substantially.

We further visualised the distribution of simulated data using Principal Component Analysis (PCA) and t-distributed Stochastic Neighbour Embedding (t-SNE), as shown in Fig. R 2. Different colours correspond to different structures distributed in low-dimension spaces. Both two methods suggest a good diversity of simulated data. The 1D signature of CSRO, z-SDM, is not too complex (like Fig. R 1), and it is hence relatively easier to reconcile the diversity of configurations using only thousands of data points as compared to 2D images.

Finally, for the Fe-Al system with the possible occurrence of CSRO, the current model can work well for these varied compositions. This is because only the peak amplitudes in the z-SDMs will change but not the peak positions. This is also expected to work because other crystal structures are not predicted in the Fe-Al system, as per the phase diagram (Fig. R 3). The same model also was applied to the Fe-Ga system to make predictions and the results match well with previous TEM work.

In short, we supplemented more details in the revised manuscript about the diversity of the simulated data, the potential biases in the dataset, and the model's generalizability.

Supplementary Table 1 Parameters for building the crystal structure library and generating the corresponding z-SDMs bank.

Note that “ σ ” represented the standard deviation of the Gaussian function.

Category	Number of z-SDMs	$\sigma_x = \sigma_y$, nm	σ_z , nm	Detect efficiency	Peak shift, nm
BCC	3000	0.2~0.8	0.02~0.05	0.2~0.7	-0.06~0.06
D03	4000	0.2~0.8	0.01~0.05	0.2~0.7	-0.06~0.06
B ₂	3000	0.2~0.8	0.03~0.06	0.2~0.7	-0.06~0.06

Fig. R 1 Simulated z-SDMs using varied lateral resolutions.

Fig. R 2 PCA and t-SNE analysis of the simulated data. The red, grey, and orange colours represent BCC, D03, and B2, respectively.

Fig. R 3 Fe-Al phase diagram (Martin Palm, Frank Stein, and Gerhard Dehm. "Iron aluminides." Annual Review of Materials Research 49 (2019): 297-326.).

Modifications: We have added a Supplementary Note to solve this concern (Page 29).

Supplementary Note 2. The generalizability of our ML-APT model and the diversity of simulated data

The generalizability of our model and data diversity are discussed in terms of the range of the parameters for simulated data, the visualisation of the distribution of simulated data, and the application of the recognition model of Fe-Al alloys on Fe-Ga alloys.

First, as listed in Supplementary Table 1, the main parameters for simulating specific structures under different APT conditions are varied across a wide range of parameters. For both the lateral and depth resolutions, the defined range includes the reported values that are encountered in experimental APT data. Worse resolutions, i.e., leading to a loss of the capability to image atomic planes, were not considered because generating featureless patterns would be meaningless for training the ML model. Our ML model generally classifies these noisy and featureless patterns as the BCC matrix to avoid bias from low-quality z-direction signals (see Supplementary Fig. 9). The simulated range of detection efficiency contains values for commercial APT apparatuses (0.35, 0.52, and 0.8) and includes worse conditions (e.g., 0.2). We even considered the effect of local peak shifts (-0.06~0.06nm) in the simulated SDMs database, which could reflect slight distortions arising from, e.g., the evolution of the projection parameters in the reconstruction model^{78,79}. We tested the larger data size via either refining the original parameter range or extending this range slightly, but the performance of the model did not change substantially.

We further visualised the distribution of simulated data using Principal Component Analysis (PCA) and t-distributed Stochastic Neighbour Embedding (t-SNE), as shown in Supplementary Fig. 12. Different colours correspond to different structures distributed in low-dimension spaces. Both two methods suggest a good diversity of simulated data. The 1D signature of CSRO, z-SDM, is not too complex, and it is hence relatively easier to reconcile the diversity of configurations using only thousands of data points as compared to 2D images.

Finally, for the Fe-Al system with the possible occurrence of CSRO, the current model can work well for these varied compositions. This is because only the peak amplitudes in the z-SDMs will change but not the peak positions. This is also expected to work because other crystal structures are not predicted in the Fe-Al system, as per the phase diagram⁸⁰. The same model also was applied to the Fe-Ga system to make predictions and the results match well with previous TEM work.

(7) Consider including a brief section discussing the limitations of the proposed approach and potential areas for future research. Address any challenges and uncertainties that might arise when applying ML models or more complex material systems or real world applications.

Response: Many thanks for the suggestions. We have added a discussion at the end.

Modifications:

Pages 13-14: The performance of ML-APT is mainly limited by the APT data quality, especially its spatial resolutions. ML-APT will hence face limitations in the accurate detection of CSRO domains with fewer than 20 APT-reconstructed atoms. In the future, modelling the atom evaporation process ^{58,59} could improve the data quality (including detection efficiency and spatial resolutions) to maybe allow for more accurate recognition of CSRO using or extending the proposed ML-APT method, as we could include in the training additional possible distortions of the features associated to CSRO. Furthermore, the present ML-APT approach requires prior knowledge of possible CSRO structures in the studied systems. Were this method to be applied to more complex alloy systems, like medium/high entropy alloys, the limited or absent prior knowledge of CSRO configuration could preclude identification and quantification.

(8) It is stated that the spatial resolution of APT prevents precise imaging of the atomic ordering process below 1 nm in radius. This limitation may hinder the detection and classification of very small CSRO domains, and it should be acknowledged in the manuscript.

Response: We have added the below modifications:

Page 13: The performance of ML-APT is mainly limited by the APT data quality, especially its spatial resolutions. ML-APT will hence face limitations in the accurate detection of CSRO domains with fewer than 20 APT-reconstructed atoms. In the future, modelling the atom evaporation process ^{58,59} could improve the data quality (including detection efficiency and spatial resolutions) to maybe allow for more accurate recognition of CSRO using or extending the proposed ML-APT method, as we could include in the training additional possible distortions of the features associated to CSRO.

(9) Lastly, the authors focus on the high accuracy and AUC values achieved by the ML model. However, it is essential to discuss potential cases of false positive and false negatives in the classification results and the impact they may have on the overall analysis.

Response: We thank the reviewer for the comment. Now, we added more ROC analysis using the confusion matrix and supplemented the discussion on the false positive rate (FPR) and false negative rate (FNR).

As listed in the confusion matrix, for the B2-CSRO, the FNR (equal to 1-true positive rate) is equal to 0 and the FPR is only 0.13, suggesting quite low possibilities of false positive and false negative cases. It is similar to the D03-CSRO. Note that the nature of the sparse distribution of CSRO in BCC leads to the imbalance of labels. This pushed us to choose the ROC and AUC, which is insensitive to the distribution of data (Fawcett, Tom. "An introduction to ROC analysis." Pattern recognition letters 27.8 (2006): 861-874).

Fig. R 4 ROC analysis of the 1D CNN and random forest algorithms obtained using 148 experimental data. **a, b** ROC curves of the 1D CNN and random forest with uncertainties, respectively, corresponding to three kinds of structures. The relevant AUC values and standard deviations are given. **c, d** Relevant confusion matrix of D03 and B2 using the CNN, respectively. TPR, TNR, FNR and FPR represent true positive rate, true negative rate, false negative rate, and false positive rate.

Modifications: We have added the discussion in the main text and revised Supplementary Fig. 4 (same as Fig. R 4):

Page 6: The false positive and false negative rates are low for D0₃-CSRO and B₂-CSRO, suggesting a limited influence on the overall analysis.

The caption of Supplementary Fig. 4: ROC analysis of the 1D CNN and random forest algorithms obtained using 148 experimental data. **a, b** ROC curves of the 1D CNN and random forest with uncertainties, respectively, corresponding to three kinds of structures. The relevant AUC values and standard deviations are given. **c, d** Relevant confusion matrix of D0₃ and B₂ using the CNN, respectively. TPR, TNR, FNR and FPR represent true positive rate, true negative rate, false negative rate, and false positive rate. The nature of the sparse distribution of CSRO in BCC leads to the imbalance of

labels. This pushed us to choose the ROC and AUC, which are insensitive to the distribution of data⁶⁷

Referee #2 (Remarks to the Author):

The manuscript under consideration is interesting in that it presents an investigation into an important yet challenging problem of identifying Chemical Short-Range Order (CSRO) in APT data. The utilization of machine learning (ML)-based techniques in the context of material characterization, analysis, and mapping is indeed a move in the right direction as it has the potential to shed new light on microstructural features that would be elusive through conventional means. Traditionally, characterizing CSRO in disordered crystalline matrices has posed formidable challenges due to the limitations of conventional techniques and ML-enhanced approaches has the potential to break these inherent resolution limits of APT and delve into the intricate sub-nanoscale features, which would address some of the issues unanswered before for instance, anomalous property changes during heat treatment of disordered crystalline alloys.

One interesting aspect of this work is the successful evidence of non-statistical B2-CSRO (FeAl) instead of the anticipated D03-CSRO (Fe₃Al) in the Fe₁₈Al alloy which demonstrates the method's efficacy in uncovering unexpected short-range ordering phenomena. The application of the proposed strategy to modified D03-CSRO detected in the Fe-Ga system, also corroborated by other microscopy techniques, further strengthens the generalizability of the approach. Having said that, I have a number of comments to make and a few questions expected to be answered which are as follows:

Response: Many thanks for your praises. We have addressed all your concerns in the following part.

(1) Given that the ML model was trained on synthetic z-SDM dataset, how can we be confident that the model also works reliably for the experimental dataset, especially when we know that the field evaporation can be noisy (and, \textit{to some minimal extent}, random) and hence the reconstruction procedure might not faithfully reproduce the exact same features found on the lost specimen? Couldn't you potentially be mistaking some reconstruction artefacts for CSRO? In short, can we trust the reconstruction algorithm to reproduce the original features or the discrepancies between the original and reconstructed does not considerably affect the results in the given problem of identifying CSRO?

Response: Many thanks for the comment. We answered this concern on reconstruction reliability in terms of the field evaporation simulation in APT, experimental parameters, and large-scale APT simulation.

First, we have discussed this issue at length in a previous paper based on simulating the field evaporation and APT data reconstruction process (<https://doi.org/10.1017/S1431927621012952>), showing that the atomic neighbourhood relationships can be maintained, even in an alloy with 5 elements but only in the depth direction. The maximum z-direction deviation can be about 0.2 nm, but most of the atoms are still within about 0~0.1 nm (See Figs. 5 (c) in a binary alloy, 7 (a) and (b) in a high-entropy alloy in <https://doi.org/10.1017/S1431927621012952>). This ensures that the z-SDMs focusing on the depth-direction signal enable to represent the signature of CSRO domains.

Second, for experiments, to minimize the deviations in z, we adopted a relatively low temperature (about 50 K) and used voltage pulsing mode to ensure that the sequence of

detection of the ions is controlled by the strength of the electrostatic field, which is the key to maintaining the high depth resolution as discussed at length in Gault et al. M&M 2020 (<https://doi.org/10.1017/S1431927620000197>). The presence of readily visible poles in Fig. 1 is a good indication that this order is maintained, as it results from the shaping of the specimen from the field evaporation process itself. Moreover, the cross-species Fe-Al elemental pair was not analysed to avoid possible biases arising from differences in evaporation fields affecting the spatial resolution (Vurpillot, F., et al. "The spatial resolution of 3D atom probe in the investigation of single-phase materials." *Ultramicroscopy* 84.3-4 (2000): 213-224).

Finally, we synthesised large-scale APT datasets by assuming that the maximum atoms shift in the lateral and depth directions are set as 5th and 1st (0.14 nm) nearest neighbour distances, respectively. This simulates the above physical-based condition. As mentioned in the manuscript, our ML model can distinguish B2 domains from the BCC matrix in terms of spatial distributions (PCC>0.9), morphology, and size distributions (PCC=0.62). It is similar to DO3 domains.

All in all, ML-APT makes optimal use of the highest-quality, near-atomic resolution of APT, combined with its high elemental analytical ability. Thereby, this enables us to precisely reveal the morphology and size distributions of multiple types of (non-)statistical CSRO, in 3D.

Modifications: We have added the discussion in the Supplementary Notes (Page 28):

Supplementary Note 1. Reconstruction quality of APT data

The reconstruction reliability of APT is discussed in terms of the field evaporation simulation in APT, experimental parameters, and large-scale APT simulation.

First, we have discussed this issue at length in a previous paper³⁴ based on simulating the field evaporation and APT data reconstruction process, showing that the atomic neighbourhood relationships can be maintained, even in an alloy with five elements but only in the depth direction. The maximum z-direction deviation can be about 0.2 nm, but most of the atoms are still within about 0~0.1 nm. This ensures that the z-SDMs focusing on the depth-direction signal enable to represent the signature of CSRO domains.

Second, for experiments, to minimize the deviations in z, we adopted a relatively low temperature (about 50 K) and used voltage pulsing mode to ensure that the sequence of detection of the ions is controlled by the strength of the electrostatic field, which is the key to maintaining the high depth resolution⁷⁶. The presence of readily visible poles in Fig. 1 is a good indication that this order is maintained, as it results from the shaping of the specimen from the field evaporation process itself. Moreover, the cross-species Fe-Al elemental pair was not analysed to avoid possible biases arising from differences in evaporation fields affecting the spatial resolution⁷⁷.

Finally, we synthesised large-scale APT datasets by assuming that the maximum atoms shift in the lateral and depth directions are set as 5th and 1st (0.14 nm) nearest neighbour distances, respectively. This simulates the above physical-based condition. Our ML model can distinguish B₂ domains from the BCC matrix in terms of spatial distributions

(PCC>0.9), morphology, and size distributions (PCC=0.62). It is similar to DO₃ domains.

All in all, ML-APT makes optimal use of the highest-quality, near-atomic resolution of APT, combined with its high elemental analytical ability. Thereby, this enables us to precisely reveal the morphology and size distributions of multiple types of (non-)statistical CSRO, in 3D.

(2) In the paragraph describing CSRO recognition model, it would be a good idea to describe input layer (number of neurons) and nature of the data being input (linking z-SDMs to the input layer).

Response: Thanks. We have added “Each continuous z-SDM was discretised using 93 data points into the input layer.”

Modifications: We have modified accordingly in the revised manuscript (Page 6):

A total of 10,000 data samples were generated (Methods, Supplementary Table 1, Supplementary Notes). **Each continuous z-SDM was discretised using 93 data points into the input layer.**

(3) It would be advisable to go through a couple of research papers I have linked below.
<https://doi.org/10.1016/j.pmatsci.2021.100854>
<https://doi.org/10.1016/j.pmatsci.2021.100854>
Short-range ordering is also a matter of significant interest in HEAs and I think it would be a good idea to have a cursory reference to these as well.

Response: We fully agree with you and CSRO is quite important in HEAs. Now we add the suggested paper and others as references.

Modifications: References on CSRO in HEAs are added in the revised manuscript:

8. Zhang, R. et al. Short-range order and its impact on the CrCoNi medium-entropy alloy. *Nature* 581, 283-287 (2020).
9. Ding, J., Yu, Q., Asta, M. & Ritchie, R. O. Tunable stacking fault energies by tailoring local chemical order in CrCoNi medium-entropy alloys. *Proc. Natl. Acad. Sci. U.S.A.* 115, 8919-8924 (2018).
10. Ding, Q. et al. Tuning element distribution, structure and properties by composition in high-entropy alloys. *Nature* 574, 223-227 (2019).
- 13. Hu, R., Jin, S. & Sha, G. Application of atom probe tomography in understanding high entropy alloys: 3D local chemical compositions in atomic scale analysis. *Prog. Mater. Sci.* 123, 100854 (2022).**

(4) Line 227 "believable" might not be the right word. Line 271, is it ground truth?

Response: Many thanks. We changed “believable” into “reliable” and “ground true” into “ground truth”.

Modifications: We have revised both in the revised manuscript:

The finding of B₂-CSRO is **reliable** in the Fe-Al alloy.

The recognition ability on D03-CSRO of the proposed method is also verified in chosen Fe-Ga alloy as a **ground truth** (Supplementary Fig. 10).

(5) Line 174: What is LCO an abbreviation of?

Response: Sorry for that. We want to say CSRO (chemical short-range order). Now two LCO words have been replaced with CSRO.

Modifications: We have revised them in the revised manuscript (Page 8):

Even in a truly random yet concentrated solid solution, some local environments similar to **CSRO** randomly form, with no specific ordering driving force. We also applied this ML-APT recognition model to a randomized dataset (Method, Fig. 2g), enabling the identification of randomly-occurring small CSRO domains. The PCC of 0.39, in this case, shows that our ML model prediction (PCC 0.67) can identify the underlying order and distinguish between **CSRO** and random structures.

REVIEWERS' COMMENTS

Reviewer #1 (Remarks to the Author):

The authors have addressed all the concerns efficiently and improved the manuscript quality drastically. I have no more concerns and recommend this manuscript for publication. I believe this work will be beneficial for the APT community.

Reviewer #2 (Remarks to the Author):

Dear Authors

Thank you for your reply. It fully covers all the questions I had.

Thank you